Reactive Transport Modeling for Supporting Climate Resilience at Groundwater Contamination Sites

Zexuan Xu[1], Rebecca Serata[1], Haruko Wainwright[1], Miles Denham[2], Sergi Molins[1], Hansell Gonzalez-Raymat[4], Konstantin Lipnikov[3], David Moulton[3], Carol Eddy-Dilek[4]
1. Lawrence Berkeley National Laboratory; 2. Panoramic Environmental Consulting; 3. Los Alamos National Laboratory; 4. Savannah River National Laboratory
Correspondence: Zexuan Xu (zexuanxu@lbl.gov)

# ABSTRACT

Climate resilience is an emerging issue at contaminated sites and hazardous waste sites, since projected climate shifts (e.g., increased/decreased precipitation) and extreme events (e.g., flooding, drought) could affect ongoing remediation or closure strategies. In this study, we develop a reactive transport model (Amanzi) for radionuclides (uranium, tritium, and others) and evaluate how different scenarios under climate change will influence the contaminant plume conditions and groundwater well concentrations. We demonstrate our approach using a two-dimensional reactive transport model for the Savannah River Site F-Area, including mineral reaction and sorption processes. Different recharge scenarios are considered by perturbing the infiltration rate from the base case, as well as considering cap failure and climate projection scenarios. We also evaluate the uranium and nitrate concentration ratios between scenarios and the base case to isolate the sorption effects with changing recharge rates. The modeling results indicate that the competing effects of dilution and remobilization significantly influence pH, thus changing the sorption of uranium. At the maximum concentration on the breakthrough curve, higher aqueous uranium concentration implies that sorption is reduced with lower pH due to remobilization. To better evaluate the climate change impacts in the future, we develop the workflow to include the downscaled CMIP5 (Coupled Model Intercomparison Project) climate projection data in the reactive transport model, and evaluate how residual contamination evolves through 2100 under four climate Representative Concentration Pathway (RCP) scenarios. The integration of climate modeling data and hydro-geochemistry models enables us to quantify the climate change impacts, assess which impacts need to be planned for, and therefore assist climate resiliency efforts and help guide site management.

# 1. INTRODUCTION

Changing climate may pose a major risk in environmental remediation, especially with regard to the fate, transport, including both hydrologic and reactive processes (Maco et al., 2018). In particular, many sites are managed with monitored natural attenuation strategies where an expanded contamination plume with high concentrations of tritium, uranium and other chemical species remain in the subsurface (Denham et al., 2020). Hydrological shift has been identified as one of the key drivers influencing such risk and uncertainty. In a changing climate,

precipitation and evapotranspiration (ET) regimes can change both in magnitude and timing, significantly affecting infiltration. Precipitation regimes are expected to change depending on where the site is located (e.g., Lambert et al., 2008). Increasing ET is usually predicted in climate model projection, due to increasing temperatures under global warming (e.g., Abtew and Melesse, 2013, Milly and Dunne, 2016). Extreme events, such as heavy rain and prolonged

droughts, are expected to become more frequent and thus may result in faster plume remobilization (e.g., Rahmstorf and Coumou, 2011).

We may define climate resilience at contaminated sites as the capacity of individual waste disposal sites to return back to the system's original condition when affected by climate trends,

climate variability, extreme events, and other climate-change-related impacts. A critical need exists for understanding climate change impacts on contaminated sites (e.g., U.S. EPA, 2014 and DOE, 2017), however, a quantitative estimation with climate change projection is still missing. Evaluating the effect of climate change on the abundance of water resources has been widely studied (e.g., Gellens and Roulin, 1998; Green et al., 2011; Middelkoop et al., 2001;

Pfister et al., 2004), however, water quality and contamination issues were less investigated (Visser et al., 2012). Most previous researches study surface water (Wilby et al., 2006; Van Vliet and Zwolsman, 2008; Van Bokhoven, 2006; Futter et al., 2009; Schiedek et al., 2007), because of the accessibility and data availability (Green et al., 2011). In the limited studies for climate change impacts on groundwater in the subsurface domain, agricultural effluents at the regional

scale are the research focus (Bloomfield et al., 2006; Futter et al., 2009; Li and Merchant, 2013; Olesen et al., 2007; Sjoeng et al., 2009; Whitehead et al., 2009; Wilby et al., 2006; Darracq et al., 2005; Destouni and Darracq, 2009; Park et al., 2010).

Recently, Libera et al. (2019) investigated the potential impact of climate change on residual

contaminants in vadose zones and groundwater, using a groundwater flow and transport model. They investigated the complex effect of precipitation and recharge shifts, leading to either dilution and remobilization of residual contaminants. Libera et al (2019) showed that the effects of dilution and remobilization on contaminant concentrations before and after changing precipitation, depending on the well locations, and that surface barrier and source zone

monitoring are critical for mitigating the impact. However, Libera et al. (2019) only simulated tritium with decay, but did not couple with a reactive transport model to simulate other chemical species, sorption, and mineral reactions. In this study, we hypothesized that increasing recharge would decrease reactive species concentration further, since increasing the volume of water in the domain would increase pH, which limits the mobility of uranium. The impact of hydrological

shifts on reactive contaminants is expected to be more complex, especially redox and pH-sensitive heavy metals. Remobilization would also be affected by additional clean infiltration water. To test those hypotheses and evaluate the impacts, process-based flow and reactive transport models that can characterize sorption and ion exchange processes are essential for quantitatively analyzing the contaminant plume and understanding climate resilience.


This study aims to evaluate the effects of climate-driven hydrological shifts on (1) reactive contaminants and (2) mineral reactions in vadose zones and groundwater. We assume that the effect of changing precipitation and temperature can be represented by perturbations/shifts in

natural recharge through the aquifer system. In our case, climate resilience is evaluated by the
concentrations of monitoring wells after climate events in comparison to the background baseline concentrations. We demonstrate our approach at the Department of Energy (DOE)'s Savannah River Site (SRS) F-Area Seepage Basins, South Carolina (SC), USA, where soil and groundwater have been contaminated by various metals and radioactive contaminants. This is the same site studied by Libera et al. (2019). The SRS F-Area seepage basin was chosen
because of the historic contamination monitoring activities, and extensive characterization and model development (e.g., Flach, 2004; Bea et al., 2013; Sassen et al., 2012; Wainwright et al., 2014, 2015, 2016; Denham and Eddy-Dilek, 2017; Libera et al., 2019). More importantly, the contamination in F-Area is representative that our study may provide broader insights to other contamination sites. The vadose zone residual contaminants are quite common (e.g., Stubbs et
al., 2009, Zachara et al., 2005) and also the contaminant export through the wetland region are fairly common features across many contaminated sites as well (e.g., Mansoor et al., 2006; Li et al., 2014, Change et al., 2014). Hence, the SRS has become a unique study site for investigating the potential consequences of climate change on contamination remobilization and mineral reactions/interactions. Our research focuses on the effect of climate change on
progress toward return to natural conditions of the plume between the basins and the funnel-and-gate. This is important to the timing of the transition of the site from enhanced to monitored natural attenuation, and hence, important to the overall effectiveness of remediation. More importantly, this work will support the risk management under changing climate conditions.

# 2. SITE DESCRIPTION

The Savannah River Site F-Area in South Carolina is approximately 100 mi (i.e., 161 km) away from the Atlantic Ocean and occupies an area of about 800 km$^2$ (Figure 1). The site was used to produce special radioactive isotopes, plutonium, and tritium, for nuclear weapons during the
Cold War Era. The F-Area is located in the north-central part of SRS. There are three hydrostratigraphic units within the Upper Three Runs Aquifer, shown in Figure 1 (B): an Upper Aquifer zone (UUTRA), a Tan Clay Confining Zone (TCCZ), and a Lower Aquifer zone (LUTRA). The UUTRA and LUTRA are mainly composed of clean sand, while the TCCZ is a low-permeability mixed sand-and-clay layer. The piezometric head measurements indicate that the
UUTRA and LUTRA units are hydrologically connected. The bottom of the LUTRA consists of a competent clay layer confining unit that separates the deeper aquifer (Gordon Aquifer) from the upper two aquifer units (Figure 1). The historical monitoring data collected at the SRS have shown that the F-Area contaminant plume migrates within the UUTRA and LUTRA (Figure 2), discharging into a local stream called Fourmile Branch Creek (FMB).

Low-level radioactive acidic waste was disposed of in three separate unlined seepage basins (F-1, F-2, and F-3) and leached into the groundwater. The basins received approximately 7.1 billion liters of waste solutions from processing irradiated uranium between 1955 and 1988. After the waste discharge operation was terminated in 1988, the F-Area basins were closed and
capped with a low-permeability material. Currently, an acidic contaminant plume extends from

the basins approximately 600 m downgradient to the groundwater seepage near the FMB. Several measures have been taken to reduce the environmental impacts at the F-Area site, including capping the basins and pump-and-treat remediation of contaminated groundwater. Since 2004, the site has been undergoing enhanced natural attenuation using a funnel-and-gate system, which consists of groundwater flow barriers to decrease the groundwater gradient, and base injection to neutralize pH and in turn immobilize uranium. The funnel-and-gate system is operating and requires the periodic injection of base solution to increase pH and immobilize contaminants. Quantitative estimation from the modeling study will provide insights for site management and stakeholders on when it is appropriate to transition the site to natural attenuation status without any treatments. Despite the many active remediation measures, the groundwater remains unnaturally acidic upgradient of the funnel-and-gate and contaminated with various radionuclides.

One characteristic of the SRS F-Area is the high acidity of the plume, making U(VI) highly mobile. The natural groundwater pH is slightly acidic, between 5.0 and 6.0, and decreases to values approaching 3.2 in the most contaminated locations. Despite many years of active remediation, contaminated groundwater still remains highly acidic, and the concentrations of U(VI) and other radionuclides are still significant (Seaman et al., 2007, Savannah River Nuclear Solution, 2021). It should be noted that in the acidic pH range at the SRS-F-Area, $K_d$ values for U(VI) could change between $10^2$ to $10^6$ (Davis et al., 2004; Dong et al., 2012). In addition, competing sorption between U(VI) and $H^+$ is important in remediation and has been well studied in the F-Area site (Davis et al., 2004, Bea et al., 2013, Arora et al., 2018). Because of the difficulty and apparent uncertainty in assessing the adsorption properties and mobility of U(VI) under complex geochemical conditions in groundwater, several researchers have performed quantification (UQ) related to U(VI) and $H^+$ competing sorption in the F-Area site (e.g., Curtis et al., 2006; Hammond et al., 2011).

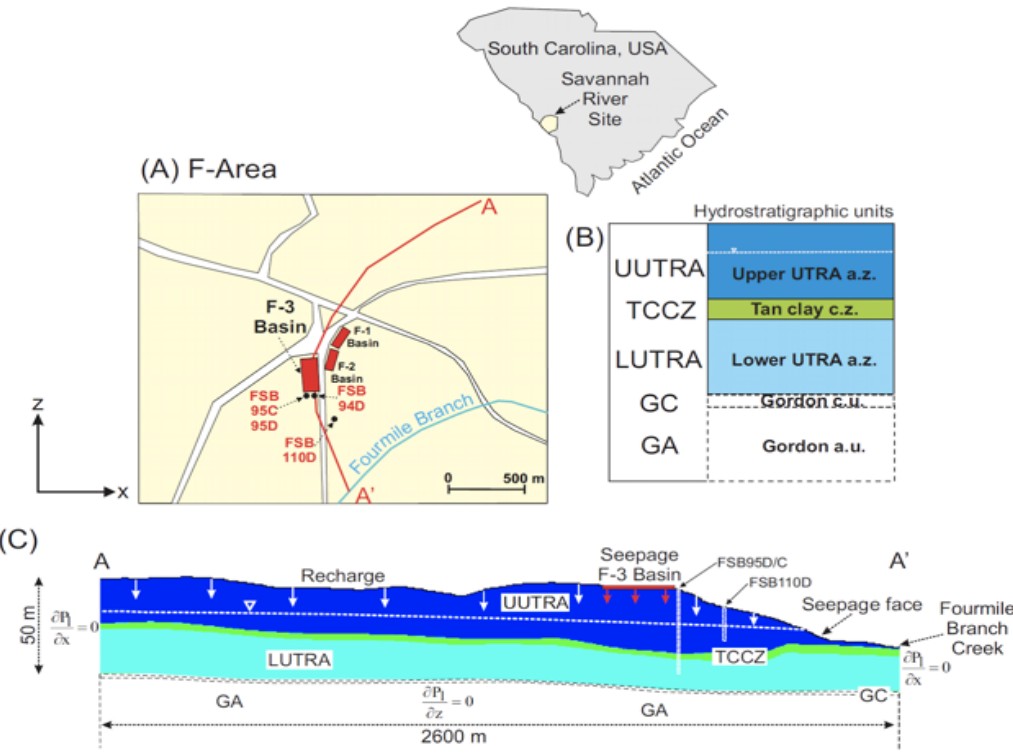

Figure 1. (A) Location of seepage basins in the F-Area of the Savannah River Site (SRS); (B) Hydrostratigraphic units defined for the F-Area; (C) 2D-cross section model domain. Modified from Bea et al. (2013).

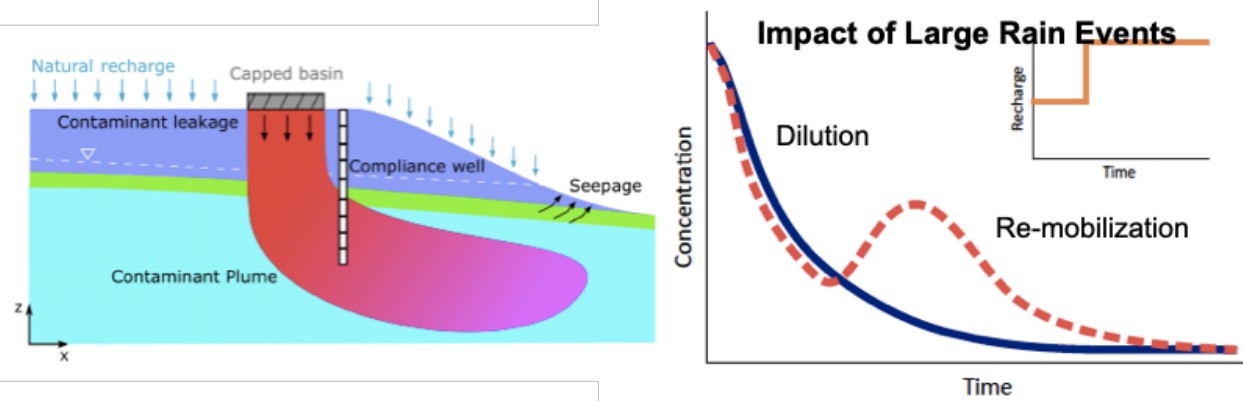

Figure 2. Left) Illustration of the hydrological conceptual model under investigation, representing a vertical two-dimensional cross section driven along the middle line of the contaminant source zone; Right) Schematic diagram of the impact of increased recharge on the concentration breakthrough curve (BTC) at an observation well located downgradient from the source zone. Modified from Libera et al. (2019).

# 3. MODELING METHODS

## 3.1. Reactive Transport Modeling with Amanzi and PFLOTRAN

Groundwater flow and contaminant transport are simulated by the numerical code Amanzi (Moulton et al., 2013, https://github.com/amanzi/amanzi), which provides a flexible and extendable parallel flow and reactive transport simulation capability for environmental applications. Amanzi has the capabilities to solve coupled unsaturated and saturated groundwater flow, as well as advection-dispersion transport equations. It includes a general
polyhedral mesh infrastructure, and provides multiple advanced nonlinear solvers with open source libraries. The reaction of contaminants and minerals carried by flow through the surrounding rock and soil is modeled by coupling with the geochemistry engine of PFLOTRAN (Lichtner et al., 2015) via the generic interface Alquimia (https://github.com/LBL-EESA/alquimia-dev). PFLOTRAN simulates the mineral reactions, adsorption, and ion exchange, while
groundwater flow and transport are simulated by Amanzi.

## 3.2. Model Setup and Boundary Conditions

The two-dimensional (2D) flow and transport Amanzi model developed in Libera et al. (2019) was employed and expanded with the coupling of reactive transport model in this study. Our Amanzi simulation used the same conditions of mineral composition and kinetic reactions as the
TOUGHREACT model in Bea et al. (2013). The 2D domain is approximately 2600 m long and 100 m deep along the groundwater flow line, passing through the middle of the F-3 basin of the SRS. Bea et al. (2013) calibrated the model and verified it using observational geochemical concentration data from several monitoring wells, and also evaluated the sensitivities of key parameters in the modeling. The model includes the vadose zone and three hydrostratigraphic
units (i.e., UUTRA, LUTRA and TCCZ) defined in the previous section. We assume homogeneous average hydrogeological properties within each unit (see Table 2), whose values are compiled from available site investigation reports. Table 1 specifies porosity, permeability, and capillary pressure/saturation data for the vadose zone (Flach et al., 2004; Bea et al., 2013). The system is considered to be advection dominated. Based on the study of scale-dependent
advection and dispersion processes in Molz (2015) that states contaminant transport will be typically dominated by advection at scale of 1000 meter, the system is considered to be advection dominated, and mechanical dispersion and molecular diffusion transport processes are neglected.

TABLE I. Physical model parameters used in the simulations. $\alpha$, n and $\Theta$r are the parameters of inverse air entry suction, a measure of the pore-size distribution, and residual water content, respectively, in the van Genuchten water retention curve.

| Hydrostratigraphic unit | Porosity [-] | Permeability [m$^2$] | $\alpha$ [-] | n [-] | $\Theta$r [-] |
|---|---|---|---|---|---|
| Upper aquifer (UUTRA) | 0.39 | $5 \times 10^{-12}$ | $4 \times 10^{-4}$ | 1.37 | 0.18 |

| | | | | |
|---|---|---|---|---|
| Tan clay (TCCZ) | 0.39 | $1.98 \times 10^{-14}$ | $5.1 \times 10^{-5}$ | 2 | 0.39 |
| Lower aquifer (LUTRA) | 0.39 | $5 \times 10^{-12}$ | $5.1 \times 10^{-5}$ | 2 | 0.41 |

No-flow boundary conditions are assigned along the two vertical sides of the 2D-cross section (see Figure 2) according to the groundwater divides, modified from previous studies (Flach, 2004; Bea et al., 2013). An impervious flow boundary condition (i.e. no-flow) is set at the bottom of the computational domain, since the confining unit at this location is highly clay-rich and continuous (Bea et al., 2013). Recharge rate is computed by the difference of climatological average precipitation and ET. This is appropriate for this domain and most groundwater models in which the groundwater domain is deep compared to the root zone depths.

The geochemical initial and boundary conditions in Table 2 are set to be the same as Bea et al. (2013), with a small modification of the nitrate-concentration initial condition for better matching with the observation. The pCO2 concentration is based on previous publications (Bea et al., 2013) and assumed constant over the simulation, as increasing pCO2 concentration has limited impacts on pH than the acidic species in the rain. Based on previous studies and field investigations, eight minerals are simulated in the reactive transport model in the F-Area. The dissolution and precipitation of initial minerals (i.e., quartz, kaolinite, and goethite) were modeled using kinetic-rate expressions derived from the literature and listed in Table 3. Gibbsite, jurbanite, basaluminite, opal, and schoepite are the species that can form when the plume interacts with the solids.

TABLE 2. Chemical composition for the background (initial), recharge and seepage solutions (modified from Bea et al., 2013). Unit is mol kgw$^{-1}$, except pH and $CO_2$ (aq).

| Mineral | Background and Recharge | Seepage |
|---|---|---|
| pH | 5.4 | 1.54 |
| $Al^{3+}$ | $2.21 \times 10^{-8}$ | $1.00 \times 10^{-8}$ |
| $Ca^{2+}$ | $1.00 \times 10^{-5}$ | $1.00 \times 10^{-5}$ |
| $Cl^-$ | $9.98 \times 10^{-3}$ [a] | $3.39 \times 10^{-4}$ |
| $Fe^{3+}$ | $2.92 \times 10^{-16}$ [b] | $2.75 \times 10^{-6}$ |
| $CO_2$ (g) | $10^{-3.5}$ [c] | $10^{-3.5}$ [c] |
| $K^+$ | $3.32 \times 10^{-5}$ | $1.72 \times 10^{-6}$ |
| $Mg^{2+}$ | $5.35 \times 10^{-3}$ | $2.47 \times 10^{-5}$ |
| $Na^+$ | $2.78 \times 10^{-4}$ | $6.82 \times 10^{-5}$ |
| $SiO_2$ (aq) | $1.77 \times 10^{-4}$ | $1.18 \times 10^{-4}$ |
| $SO_4^{2-}$ | $2.25 \times 10^{-5}$ | $4.80 \times 10^{-5}$ |

| | | |
|---|---|---|
| Tritium | $1.0 \times 10^{-15}$ | $2.17 \times 10^{-9}$ |
| $NO_3^-$ | $1.0 \times 10^{-4}$ | $1.00 \times 10^{-2}$ |
| $UO_2^{2+}$ | $1.25 \times 10^{-10}$ | $3.01 \times 10^{-5}$ |

a: Calculated as electric charge balance; b: Equilibrium with Kaolinite; c: fixed by atmosphere pressure of

TABLE 3. Initial mineral volumetric fraction distribution in the simulation (Bea et al., 2013).

| Mineral | wt.% [-] | Vol. frac. [-] | Surface area [$m^2 g^{-1}$] | Density [g $cm^{-3}$] |
|---|---|---|---|---|
| Quartz | 94.5 | 0.9496 | 0.14 | 2.648 |
| Kaolinite | 4.015 | 0.0412 | 20.71 | 2.594 |
| Goethite | 1.485 | 0.0093 | 16.22 | 4.268 |
| Schoepite | 0 | 0 | 0.1 | 4.874 |
| Gibbsite | 0 | 0 | 0.1 | 2.44 |
| Basaluminite | 0 | 0 | 0.1 | 2.119 |
| Opal | 0 | 0 | 0.1 | 2.072 |
| Jurbanite | 0 | 0 | 0.1 | 1.789 |

While Bea et al. (2013) implemented an electrostatic sorption model developed previously by Dong et al. (2012), which is less numerically efficient and requires additional parameterization. Arora et al. (2018) developed a non-electrostatic sorption model (NEM) at the F-Area site, and demonstrated that NEM achieved the same predictive performance as a surface complexation model (SCM) with electrostatic correction terms. The SCM approach is computationally expensive and requires the estimation of additional parameters that describe mineral surface characteristics. On the other hand, NEM does not consider the effects of the development of surface charge on the formation of surface complexes, and it also simplifies the parameters needed in the reactive transport modeling. In Arora et al. (2018), three mineral surface sites with different site densities and acidity constants are developed for modeling $H^+$ sorption and transport, then further extended to noncompetitive and competitive $H^+$ and U(VI). In this paper, we use the competitive $H^+$ and U(VI) sorption NEM parameters (including site density and surface complexation constant listed in Table 4), which are derived from an inverse analysis and calibration by Arora et al. (2018), and implement them in the model.

TABLE 4. NEM model parameters for $H^+$ and U(VI) competitive sorption (Arora et al., 2018).

| Site | Site density (moles/$m^2$) |
|---|---|
| >TOH | $7.0 \times 10^{-7}$ |

| | |
|---|---|
| >XOH | $1.6 \times 10^{-6}$ |
| >YOH | $9.0 \times 10^{-7}$ |

| Reactions | Surface Complexation Log K |
|---|---|
| $>TOH_2^+ \;---\; >TOH + H^+$ | -4.77 |
| $>TO^- \;---\; >TOH^- + H^+$ | 4.73 |
| $>XOUO_2^+ \;---\; >XOH + UO_2^{2+} - H^+$ | -0.67 |
| $>YOH_2^+ \;---\; >YOH + H^+$ | -3.41 |

## 3.3. CMIP5 Climate Scenarios

CMIP5 (Coupled Model Intercomparison Project, Taylor et al. (2012) is an experimental protocol with an ensemble of global climate model outputs to improve understanding of climate, and to
provide estimates of future climate change that will be useful to those considering its possible consequences. The climate forcing in our study used the 1/8-degree downscaled CMIP5 outputs at the F-Area study site from January 1950 to December 2100. The ensemble outputs include 28 models with four climate scenarios (RCP2.6, 4.5, 6.0 and 8.5) in the future climate projection. The top soil at the F-Area study site is sandy (Wainwright et al., 2014), so we
assume that surface runoff is negligible. In other words, infiltration is calculated by subtracting evapotranspiration (ET) from precipitation, which are simulated by the atmospheric and land surface models, respectively, from the coupled climate models. The figure below shows that the 10-year moving average of selected variables demonstrates that both precipitation and ET have increased approximately 6% since the 1950s to the present, and will keep increasing up to an
additional 6% by the end of this century. The differences among climate scenarios are not statistically significant, but the highest greenhouse gas concentrations (i.e., RCP8.5) ensemble simulates higher precipitation and ET than others. Although total recharge only slightly increases as both precipitation and ET are increasing (hence the difference offsets), our simulations focus on gaining the understanding and quantitative estimation of changing climate
impacts on the long-term robustness of contamination remediation in the F-Area.

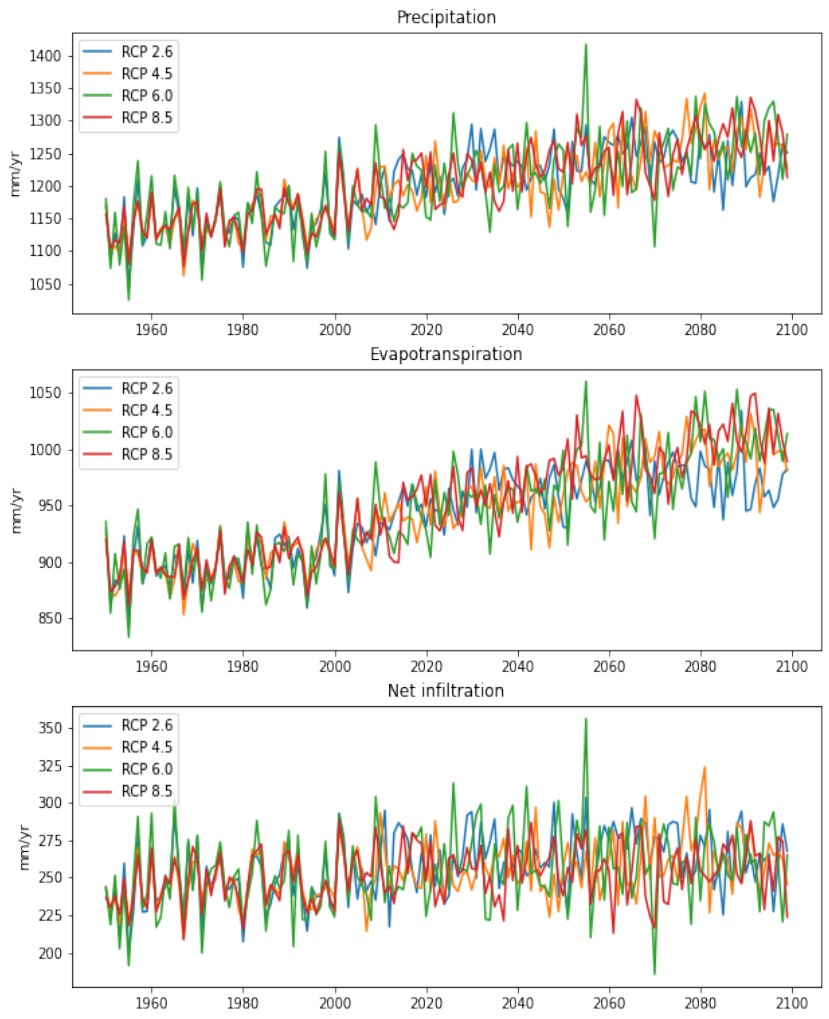

Figure 3. Simulated precipitation, evapotranspiration, and net infiltration (precipitation - evapotranspiration) at different climate projection scenarios from the CMIP5 datasets.


## 3.4. Modeling Scenarios

The modeling scenarios were developed based on (Libera et al., 2019) are only briefly described here. The modeling scenarios cover two stages of the F-Area historical operation and one additional stage in the future projection. The waste disposal was active during the period 1955-1988, and the basins were capped in 1988, when seepage from the basins into the


vadose zone is assumed to have stopped. This study evaluates the effect of climate-change-induced variations in recharge on contaminant transport after 2020. A base case was developed with a constant recharge rate throughout the simulation period for assessing climate change impacts. The uniform recharge rate is 4.743 x $10^{-6}$ kg-water/m$^2$/sec (0.15 m/yr infiltration rate), based on the estimation in Bea et al. (2013). Furthermore, we developed three perturbed recharge scenarios with respect to the baseline recharge conditions. The three scenarios are: (1) constant positive recharge shift from 2020, i.e., increasing precipitation scenarios; (2) constant negative recharge shift from 2020, i.e., decreasing precipitation scenarios; and (3) cap failure and constant positive case from 2020. In both increasing and decreasing scenarios, recharge changes 10%-50% after 2020. we hypothesize a complete failure of the containment structure scenario, which is represented by increased source-zone recharge of 10%-50% to the level of the surrounding region. Multiple studies have demonstrated that increased vegetation or other mechanisms could threaten or completely damage the integrity of the source-zone capping structure (Worthy et al., 2013, 2015). In addition to the perturbation scenarios, the contaminant transport and plume remobilization simulated by Amanzi are also forced by the four projection scenarios of CMIP5 ensemble climate model data, i.e., climate model scenarios. Instead of the constant recharge rate in those scenarios with changing precipitation, the annual recharge rate is used in CMIP5 climate scenarios from 1950 to 2100.

# 4. RESULTS

## 4.1. Base Case

The plume migration is depicted in Figure 4 for the base-case results described in Bea et al. (2013). The plume migrates through the vadose zone and then infiltrates vertically downward until it reaches the groundwater table (Figure 4a). The plume then migrates vertically through the TCCZ into the LUTRA, and also horizontally downstream closer to the FMB (Figure 4b). Despite the low permeability of the TCCZ, leakage from the UUTRA to the LUTRA is observed over most of the flow domain. After basin closure and capping, the seepage from the basin is assumed to stop. The uncontaminated groundwater arriving from upgradient increased pH and reduced the U(VI) concentration (Figure 4c). After the basin closure, because the vadose zone flow stops, pH remains low and U(VI) concentrations high in the vadose zone. In addition, the uranium concentration is higher in the TCCZ, where the permeability is low. The vadose zone below the basin appears to act as a long-term contaminant source for groundwater in the deeper layers (Figure 4d). Although aqueous uranium concentration decreased by several orders of magnitude after the basin was capped, it is still higher than background concentration.

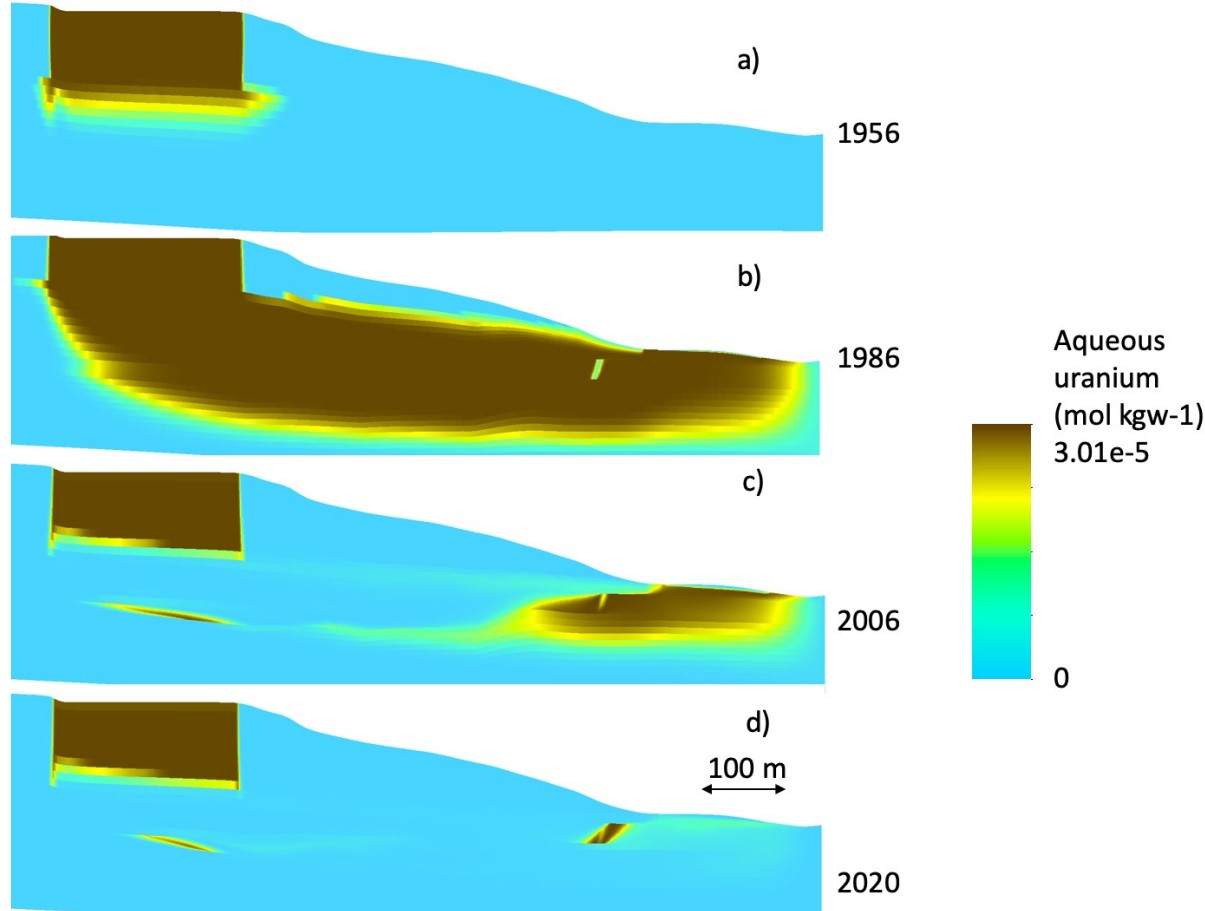

Figure 4. Plume profile of aqueous uranium concentration in the downstream of F-Area study site from 1954 to 2020 in the base case simulation (The vertical exaggeration is x5).


Figure 5 shows the base-case breakthrough curves of pH, aqueous uranium, tritium, and nitrate at the source-zone well (FSB-95DR) and the downgradient well (FSB-110D) for the full simulation period (1956-2100). Both wells are located in the UUTRA layer. The simulated pH values rapidly decrease to 3.3 at both the source-zone well (Figure 5a) and the downgradient

well (Figure 5c). In general, tritium concentrations (Figure 5c) decrease faster and more dramatically than aqueous uranium and nitrate, owing to its radioactive decay. The uranium concentrations (Figure 5b) increased from the background level $1.25 \times 10^{-10}$ mol kgw$^{-1}$ to $3.0 \times 10^{-5}$ mol kgw$^{-1}$ at both wells in less than a few years, and remained constant until basin closure in 1988. After the basin closure, pH rebounds to 4.0 in 2000 and gradually increases throughout

the end of simulation. Similarly, uranium concentration (Figure 5b) decreases by two orders of magnitude in 20 years and keeps decreasing to approximately $1.0 \times 10^{-7}$ mol kgw$^{-1}$ by the end of the simulation period. Compared to the downgradient well, the source-zone well consistently has lower pH (Figure 5a) and higher aqueous uranium (Figure 5c) concentrations throughout the simulation period. By the end of 2100, pH (Figure 5a) is higher than 5.0 and aqueous

uranium concentration lower than $2 \times 10^{-7}$ mol kgw$^{-1}$ (Figure 5b) in most of the vadose zone at the source-zone well. Overall, our simulation has a similar fit to the observations as Bea et al (2013) as the same parameters in Bea et al (2013) are used.

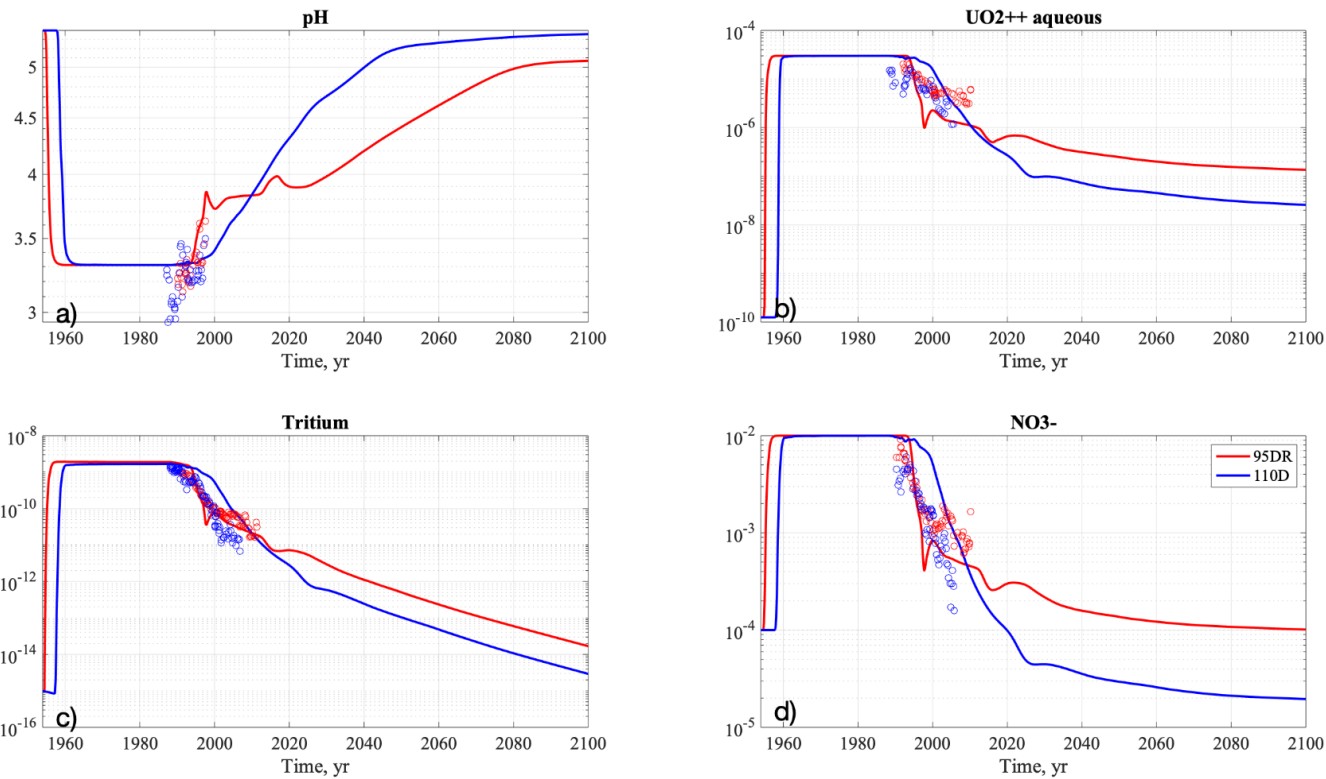

Figure 5. Breakthrough curves of pH, aqueous uranium, tritium and nitrate at the source-zone well and downgradient well in the base case over the simulation period (1954-2100).

## 4.2. Increasing Recharge Scenarios

The breakthrough curves under the increasing recharge scenarios are shown in Figure 6. When recharge is increased, pH at the source-zone well (Figure 6a) is significantly lower compared to the base-case scenario. pH values are changed with different recharge rates as relatively high pH infiltrated rainwater dilutes the low-pH contaminated environment in the subsurface system. However, the relationship between recharge and pH is nonlinear, with thresholds such that pH is the lowest at +20% recharge, while pH is higher in the cases with +30% to +50% recharge. Nitrate concentrations at the source-zone well (Figure 6b) increase immediately after 2020, and spike 5 years after perturbation, with the highest concentration in the greater recharge (+50%) case. After 2050, nitrate concentration is the highest with +20% recharge, and decreases from +30% to +50% recharge (Figure 6b). The tritium concentrations (Figure 6c) peak similarly to nitrate, although tritium decreases significantly after 2040 due to radioactive decay. The uranium concentrations (Figure 6d) are also similar to the breakthrough curves for the nitrate concentrations and show negative correlation with pH oscillation. At downgradient locations, pH (Figure 6e) is not influenced by the recharge increase up to +30%. Above the 40% increase, pH




decreases significantly after 2040. Nitrate concentrations at the downgradient well (Figure 6f)
decrease immediately after 2020 due to dilution, but increase afterwards, with peaks around
      2040. Similar to the source-zone well (Figure 6b), the concentration peaks are higher with
      greater recharge rates and remain higher than the base case throughout the end of the
      simulations. The tritium concentrations (Figure 6g) keep decreasing after 2020 with the peaks in
      2040, with the similar behavior of sudden increase showing in nitrate concentration (Figure 6f),
and higher concentrations in the high recharge scenarios. The uranium concentrations (Figure
      6h) also exhibit patterns similar to those of nitrate (Figure 6f), in that both the peak and
      remaining concentrations are higher in the greater recharge scenarios.

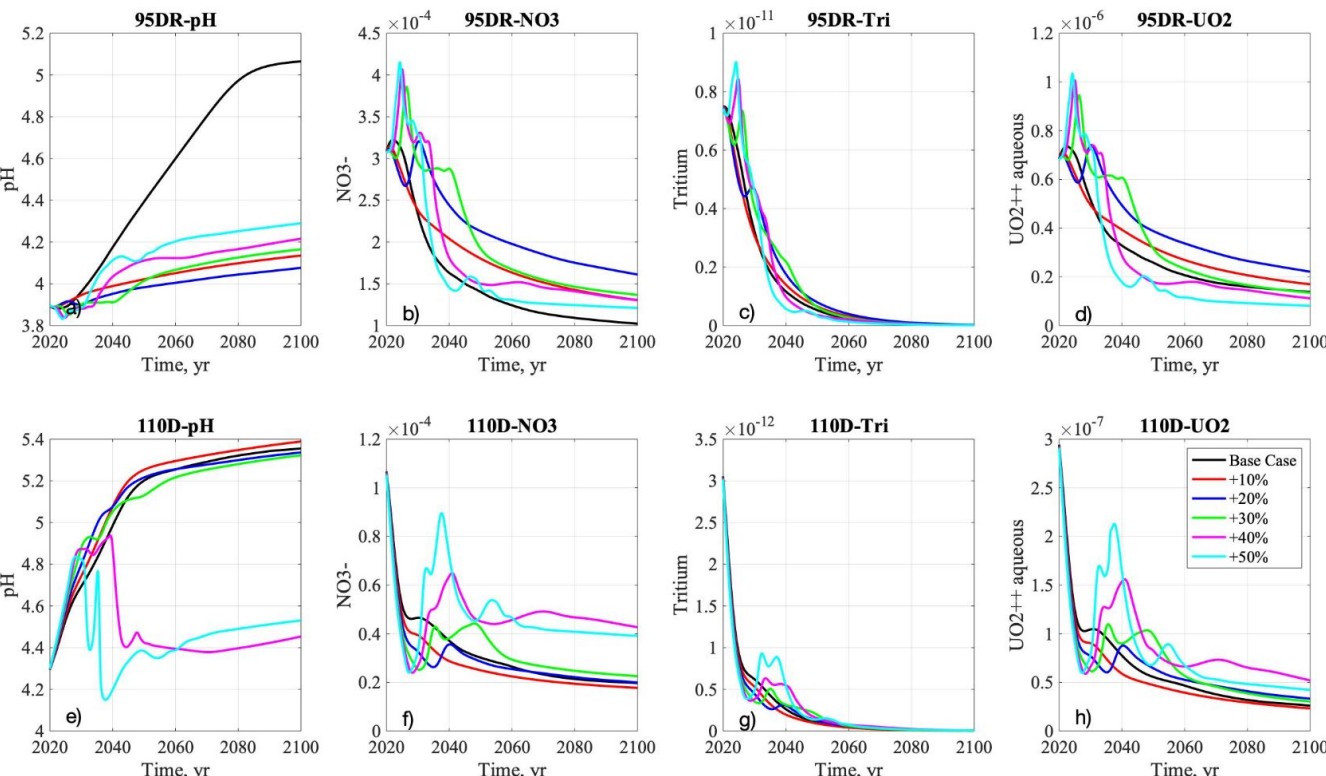

Figure 6: Breakthrough curves of pH, nitrate, tritium, and aqueous uranium at the source-zone
      well (a-d) and downgradient well (e-h) in the base case and increasing precipitation scenarios
      from 2020 to 2100.

      The reactive (uranium) and non-reactive (nitrate) species are compared in Figure 7. $K_d$ values
are computed by sorbed uranium concentration in the solid phase with the aqueous uranium
      concentration from the model outputs. Figure 7a shows that $K_d$ values at the source-zone well
      are lower in the increasing recharge cases than the base case, which is consistent with pH
      breakthrough curves (Figure 6a). The +20% case has the lowest $K_d$ at the source-zone well,
      while the $K_d$ values are higher in the smaller recharge case (+10%) and greater recharge cases
beyond +30%. In contrast, at the downgradient well (Figure 7d), the $K_d$ values are lower in the
      +40% to +50% scenarios echoing the downgradient pH breakthrough curves in Figure 6e. In

addition, we compare uranium and nitrate concentrations with respect to the maximum
concentration (i.e., the peak concentrations that occur after a few years in the increasing
recharge scenarios) as well as the average concentration from 2040 to 2100, which illustrates
the long-term contamination trend. Figure 7 (b-c) presents the ratio of uranium and nitrate,
defined as the concentration in each scenario compared to the baseline case. In the maximum
concentration at the source-zone well (Figure 7b), the ratios are mostly higher than 1,
demonstrating that the maximum concentration is higher in the greater increasing recharge
scenarios. The uranium maximum concentration ratio is higher than the nitrate; therefore, the
increasing recharge affects the uranium concentrations more than the nitrate concentrations at
the peaks (Figure 7b). For average concentrations at the source-zone well (Figure 7c), the ratio
increases in the +20% recharge case, but decreases at greater recharge values. Different from
the maximum concentrations, the mean uranium ratio becomes lower than the mean nitrate
ratio, and falls below 1.0 in the greater recharge scenarios (Figure 7c). At the downgradient
well, the maximum concentration ratios are less than 1.0 in the (+10% ~ +30%) recharge
scenarios but higher than the base case in greater recharge (+40~50%) scenarios, while nitrate
and uranium ratios are similar (Figure 7e). The average concentration ratios at the
downgradient well after 2040 are generally higher with increasing recharge, and reach their
highest at the +40% scenario (Figure 7f). The nitrate concentration ratios are lower than
uranium in the smaller (+10% ~ +30%) recharge scenarios, but are higher in those scenarios of
above +40% recharge.

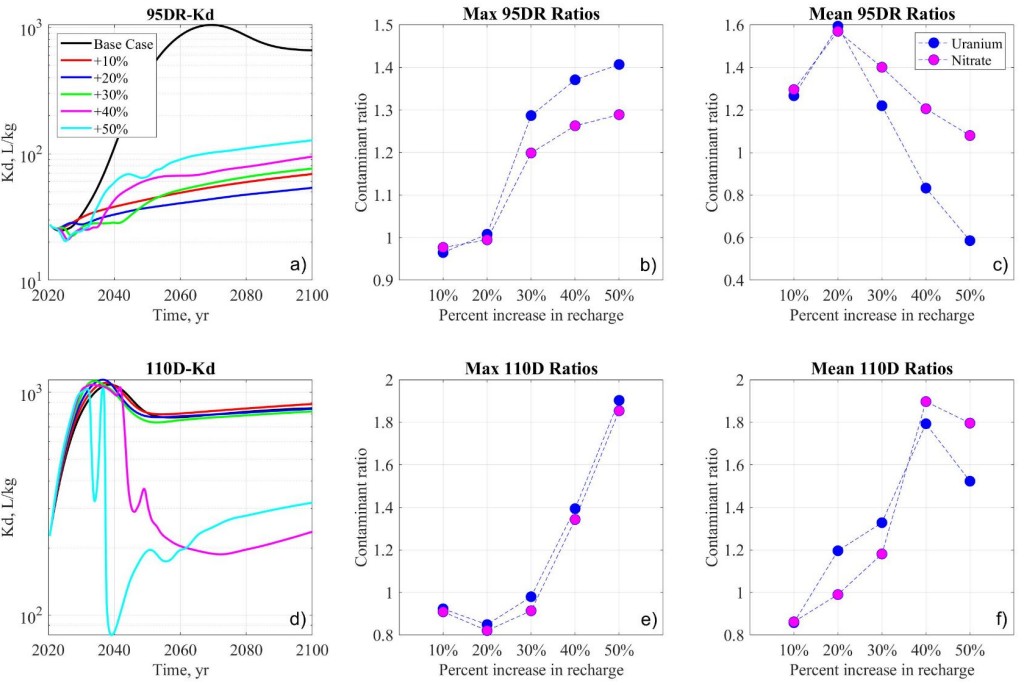

Figure 7: Breakthrough curves for $K_d$ at the source-zone well (a-c) and the downgradient well (d-
f) for the increasing recharge scenario from 2020 to 2100. Maximum and average ratios of base
case to increased recharge case for uranium and nitrate concentrations at both well locations.

## 4.3. Decreasing Recharge Scenarios






Although decreasing recharge has little impact on pH at the source-zone well up to -30% (Figure 8), pH increases significantly in the -40% ~ -50% recharge scenarios. The nitrate concentrations (Figure 8b) increase immediately after the perturbation of recharge, then decrease throughout the end of the simulation. Similar to pH, nitrate concentrations (Figure 8b) do not change significantly in smaller decreasing recharge scenarios, but decrease two orders of magnitude in the greater (-40 ~ -50%) decreasing recharge scenario. Tritium concentrations (Figure 8c) also increase immediately after 2020, then decrease; the rate of decrease is more rapid than the nitrate concentrations due to radioactive decay, and exhibit few differences among decreasing recharge scenarios. The uranium concentration (Figure 8d) breakthrough curves are similar to the nitrate curves. At the downgradient well, the pH values have a similar trend to the source-zone well in all decreasing recharge scenarios before 2040. However, the breakthrough curves diverge after 2040 and increase more in the greater decreasing recharge scenarios. The pH values are higher than the source-zone well and reach as high as 7.0 in the -50% recharge scenario in 2100 (Figure 8e). The nitrate concentrations in the down gradient well (Figure 8f) keep decreasing in the first 10-15 years after 2020. Concentrations peak around 2025-2035; the decrease is more significant in all the decreasing recharge scenarios than the base case. In general, the peak concentrations occur earlier and higher in the greater decreasing recharge scenarios, and the breakthrough curves decrease faster and lower in the long-term projection to 2100. Spikes were observed in the tritium concentration breakthrough curves (Figure 8g), as well as with smaller magnitudes at the down gradient well 10-15 years after the perturbation. The uranium concentration breakthrough curves (Figure 8h) are similar to the nitrate, but decrease more rapidly in all cases.

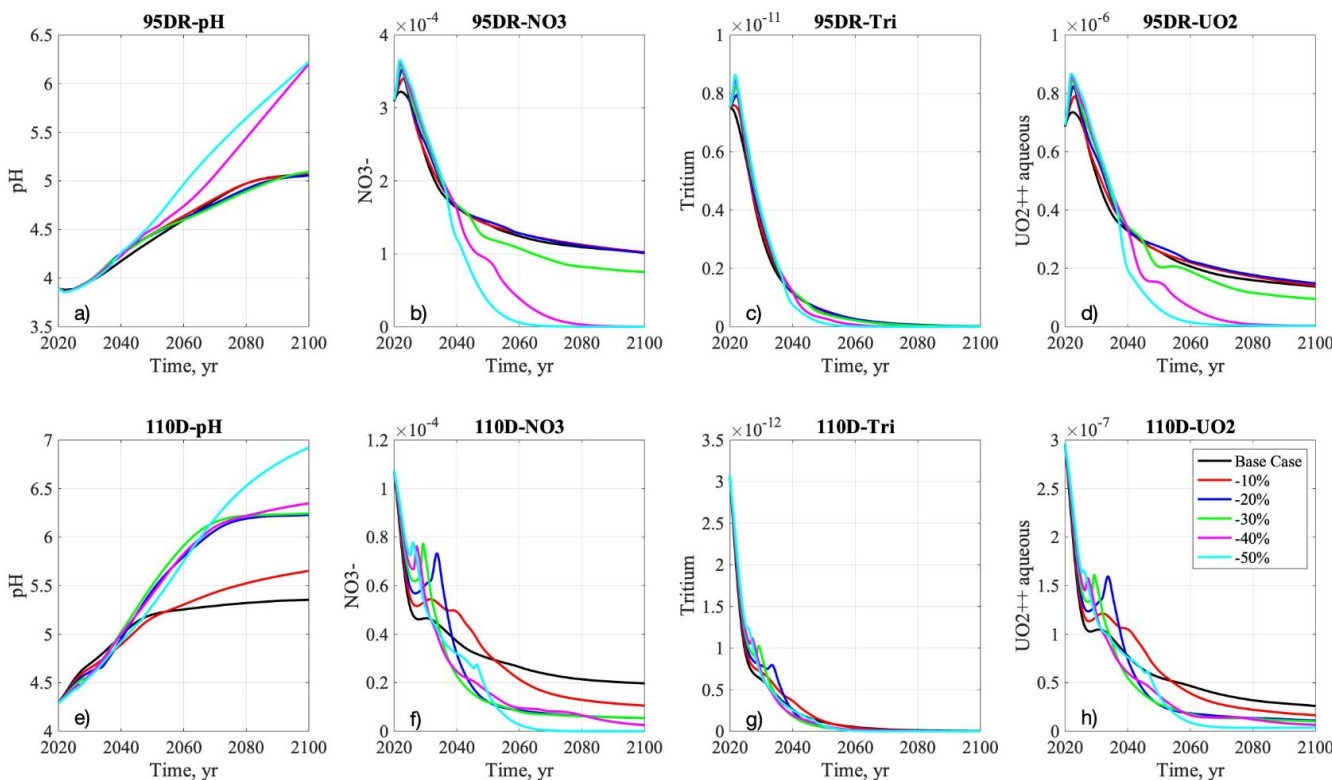

Figure 8: Breakthrough curves of pH, nitrate, tritium, and aqueous uranium at the source-zone well (a-d) and downgradient well (e-h) in the base case and decreasing precipitation scenarios from 2020 to 2100.


$K_d$ breakthrough curves generally reflect the pH breakthrough curves in Figure 8 and are higher in the decreasing recharge scenarios at both well locations. Figure 9a shows that at the source-zone well, the base case ~ -30% cases have relatively similar $K_d$ values throughout the simulation period. After 2060, the -40% and -50% recharge scenarios both significantly
increase. At the downgradient, $K_d$ values are generally higher than source-zone well, and the difference in $K_d$ values among cases are more pronounced (Figure 9d). $K_d$ value is the lowest in the higher recharge (base case and -10%) scenarios, and largest in the significantly decreasing recharge (-50%) case. The -10% and -20% recharge scenarios significantly diverge at 2040 and converge at 2100. Similar to the increasing recharge scenarios, maximum uranium and nitrate
concentrations at the source-zone well occur immediately a few years after the perturbation (Figure 8). With decreasing recharge from -10% to -50%, maximum concentration ratios are higher than 1.0 and increase with decreasing recharge, while average concentration ratios are generally lower than 1.0. In Figure 9b, the uranium maximum concentration ratios are slightly higher than nitrate, with greater difference in the -50% recharge case. In Figure 9c, the ratios of
long-term average concentrations show that both uranium and nitrate concentration are nearly the same as the base case in smaller decreasing recharge scenarios (-10% ~ -20%), but decrease quickly and are significantly lower in the greater decreasing recharge scenarios (-30% ~ -50%). Compared to the results at the source-zone well, the maximum and average

concentration ratios at the downgradient well (Figure 9e and f) have similar trends. Nitrate
maximum concentration ratios are higher than nitrate (Figure 9e), and their differences are the
greatest in the -20% and -30% recharge case. The average concentration ratios (Figure 9f)
decrease with decreasing recharges, and uranium ratios are consistently higher than nitrate.

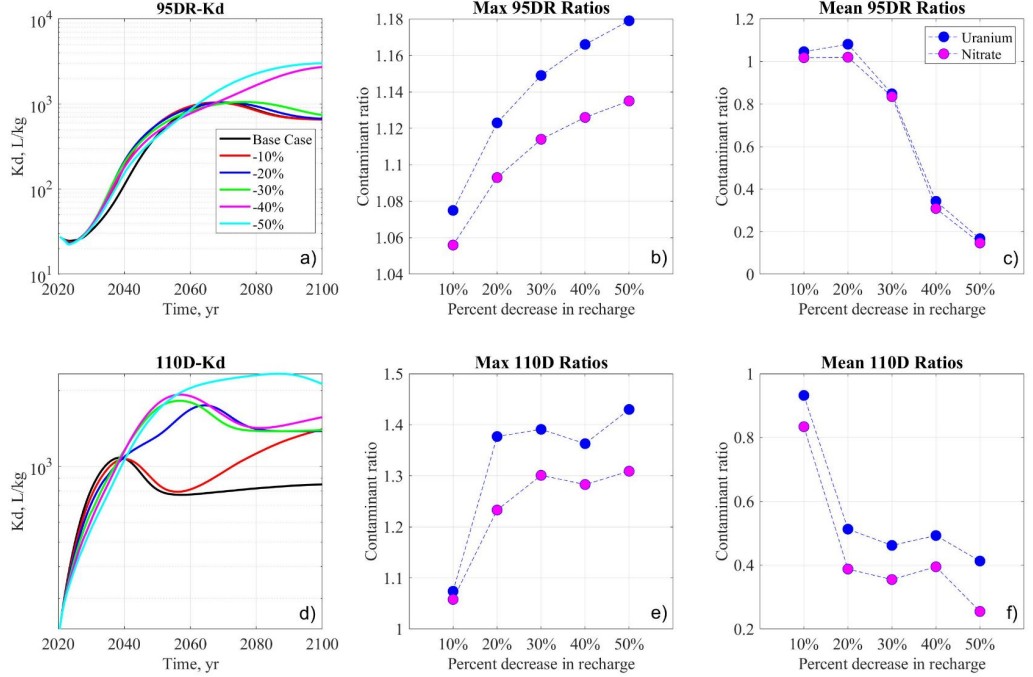

Figure 9: Breakthrough curves for $K_d$ at the source-zone well (a-c) and the downgradient well (d-
f) for the increasing recharge scenario from 2020 to 2100. Maximum and average ratios of base
case to increased recharge case for uranium and nitrate concentrations at both well locations.

## 4.4. Cap-Failure Scenarios

In the cap-failure scenarios, pH is always lower than the base case across +10% to +50%
recharge rates (Figure 10a). At the source-zone well, these pH values dip below 3.5 in 2030,
rebound to 4.0 after 2045, and then slightly increase to 4.3 by the end of the simulation. The
+50% cap-failure scenario has the highest pH value compared to the +10 ~ +40% cap-failure
cases. Nitrate concentrations spike and increase one order of magnitude in 2030, then
decrease to the same level as the base case in 2050 (Figure 10b). The pattern of tritium and
uranium breakthrough curves (Figure 10 c-d) look very similar to nitrate. Among the
breakthrough curves of nitrate, tritium, and uranium across all cap-failure scenarios, the +50%
cap-failure scenario simulates the earliest peak, while the +20% scenario simulates the highest
peak. At the downgradient well, pH values at all cap-failure scenarios increase with the base
case in the first ten years, then decrease around 2035 and remain lower than the base case
(Figure 10e). pH values only decrease from 5.5 to 5.0 in the smaller +10 ~ +20% recharge
rates. However, they decrease significantly with greater recharge rates (+30% ~ +50%) in those
cap-failure scenarios. The breakthrough curves of pH increase in the first ten years after 2020,
dip to 3.6 around 2040, and then slightly increase, but require several decades to rebound to

the same pH level as in 2020. Compared to the nitrate concentration breakthrough curves at the source-zone well (Figure 10b), the peaks at the downgradient well are simulated in 2040 with a 10-year delay (Figure 10f). The nitrate concentrations in those greater (+30% ~ 50%) recharge rates occur earlier and are higher than in the smaller (+10% ~ 20%) recharge rates. The tritium concentration shows similar peaks as nitrate, but the earliest peak with +50% cap failure has the highest values, and the later peaks with smaller (+10% ~ 20%) recharge rates will be lower because of tritium radioactive decay (Figure 10g). Uranium concentration breakthrough curves show similar behaviors to nitrate in both wells (Figure 10bd and fh).

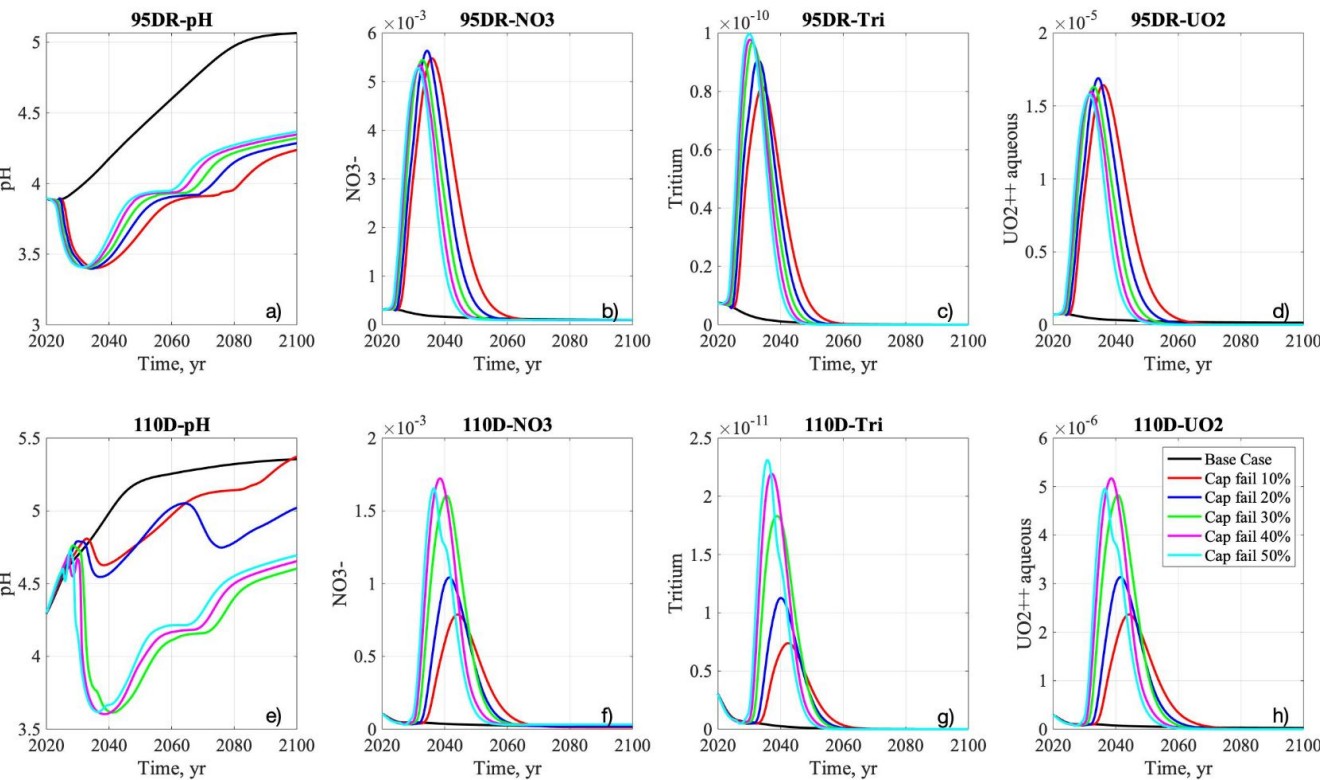

Figure 10. Breakthrough curves of pH, aqueous uranium, tritium, and nitrate at the source-zone well (a-d) and downgradient well (e-h) in the base case and cap-failure scenarios from 2020 to 2100.

Kd breakthrough curves are highly correlated with pH at both monitoring wells (Figure 10ae), $K_d$ decreases by 2035 and 2040 at both wells, respectively, returns to the 2020 level around 2050, then keeps increasing until 2100. At the source-zone well, $K_d$ values decrease and rebound fastest in the +50% recharge case, and the smallest +10% recharge rate case shows a similar trend but is delayed by nearly 10 years (Figure 11a). At the downgradient well, the $K_d$ breakthrough curves at higher recharge cases (+30% ~ +50%) are more closely correlated with pH and decrease around 2040, while smaller recharge cases (+10% ~ +20%) are more similar to the base case (Figure 11d). A turning point occurs in 2040, when the +30% case switches places with the +50% case and has the lowest $K_d$ value until 2100, similar to the behavior of

aqueous uranium breakthrough curves in 2040 (Figure 10h). When comparing Figure 11d and Figure 10e, it is clear that although pH is not the highest in the +20% cap-failure scenario, after 2070, that scenario has the highest $K_d$ value and more adsorption. In cap failure scenarios, the maximum and average uranium concentration ratios are consistently greater than nitrate in both
wells, and follow the same trend with increasing recharge rates (Figure 11b-c, e-f). Both ratios of uranium and nitrate maximum and average concentration are one order of magnitude greater than the base case. The maximum concentrations of uranium and nitrate are observed in 2030 and 2040 at the source-zone well and downgradient well, respectively (Figure 10bf), although it is difficult to tell the difference from the breakthrough curves because of the magnitude of peak
concentrations. The uranium and nitrate maximum concentration ratios are highest in the 20% cap-failure scenarios (Figure 11b), and decrease with greater increasing recharge rate. The ratios of uranium average concentrations against base case are also persistently higher than nitrate in the long term throughout 2100, and decrease with greater recharge rate (Figure 11c). At the downgradient well, the maximum concentration ratio against the base case generally
increases with greater recharge rate, and is the largest in the +40% recharge case (Figure 11e). The average concentration ratio increases with the smaller (+10% ~ +30%) recharge rates, then decreases with the greater (+40% ~ +50%) recharge rates (Figure 11f).

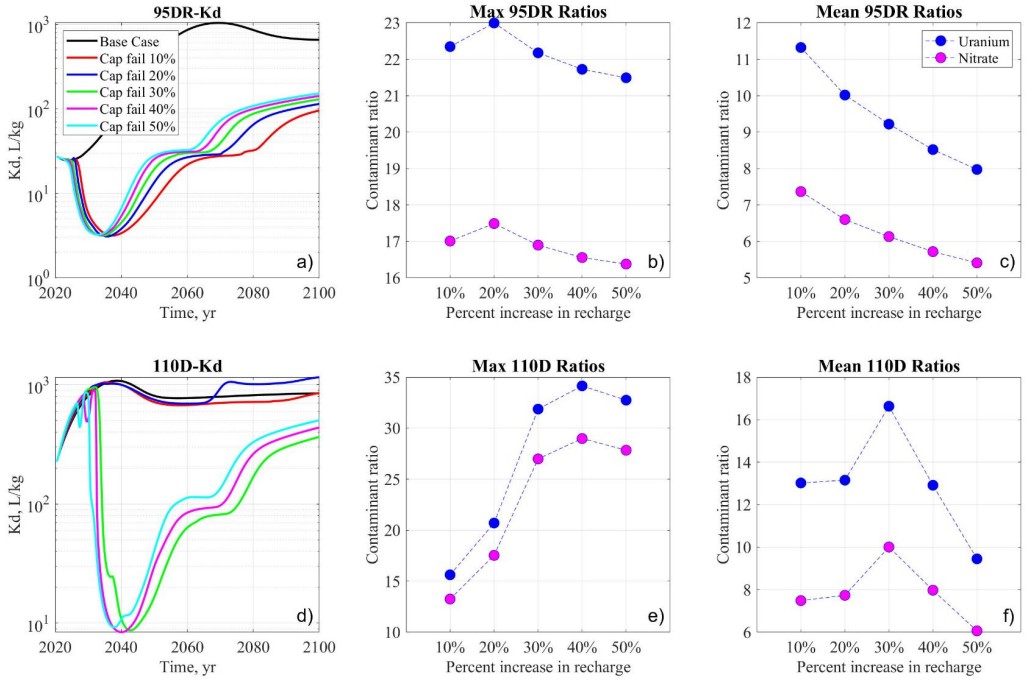

Figure 11: Breakthrough curves for $K_d$ at the source-zone well (a-c) and the downgradient well
(d-f) for the increasing recharge scenario from 2020 to 2100. Maximum and average ratios of base case to increased recharge case for uranium and nitrate concentrations at both well locations.

## 4.5. Climate Model Scenarios

Recharge rates are calculated by subtracting evapotranspiration (ET) from precipitation in the four CMIP5 climate projection scenarios. The highest greenhouse gas concentration pathway RCP8.5 scenario has the maximum simulated precipitation and evapotranspiration. However, the differences in recharge rate are small across those four scenarios as both precipitation and ET increase in the projection (Figure 3). Therefore, the concentration breakthrough curves are

similar under those climate scenarios. The average recharge rate in those scenarios is around $8.0 \times 10^{-6}$ kg-water/m$^2$/sec (0.253 m/yr), or approximately 1.68 times higher than the base case. In general, simulated contaminant concentrations in those climate scenarios are lower than the base case due to dilution effects with greater recharge rate, except that pH values are also lower than the base case (Figure 12). The breakthrough curves decrease faster before 2020

(not shown in Figure 12) and reach background concentration sooner than the base case.

At the source-zone well, the pH breakthrough curve gradually rebounds from 4.0 to 4.5 by the end of the simulation (Figure 12a). Both nitrate and uranium concentrations show annual variability after 2020, as recharge rates are changing annually (Figure 12bd). Specifically, nitrate

breakthrough curves (Figure 12b) become steady state sooner than the uranium, as nitrate background concentration is higher. The oscillation is hardly observed in tritium concentration breakthrough curves, as it decreases faster due to decay. At the downgradient well, pH values across climate scenarios are consistently lower than the base case with annual variability (Figure 12e). Compared to the results at the source-zone well, the nitrate concentrations at the

downgradient well (Figure 12f) are lower than the background level with greater annual variability, and become steady state a few years later. The tritium concentration becomes extremely low below $1.0 \times 10^{-15}$ mol kgw$^{-1}$ (Figure 12g), while uranium concentrations return to background level after 2030 (Figure 12h).

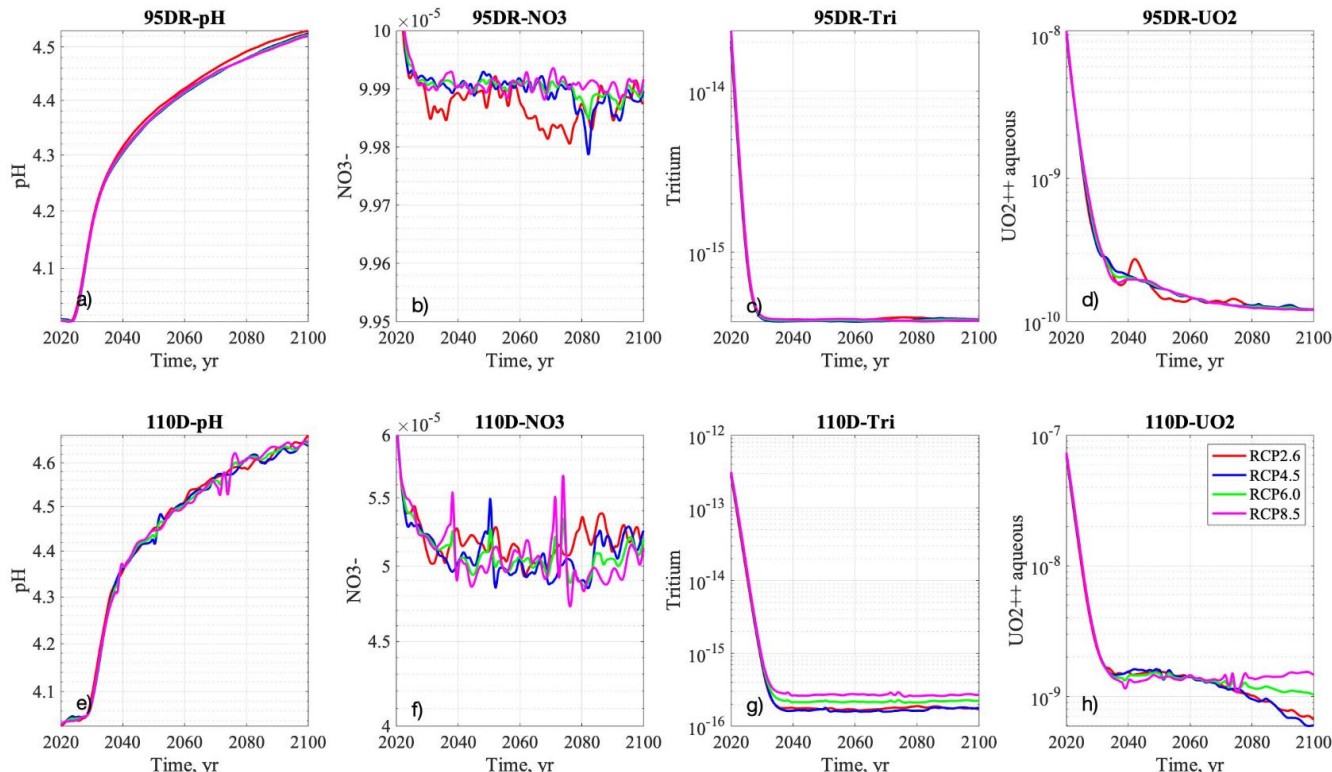

Figure 12. Breakthrough curves of pH, nitrate, tritium, and aqueous uranium at the source-zone well (a-d) and downgradient well (e-h) in the climate scenarios from 2020 to 2100.

# 5. DISCUSSION

A balance between dilution and remobilization is a key factor determining the contaminant concentration depending on recharge rates, as discussed in Libera et al. (2019). In the increasing recharge scenarios, contaminant concentrations decrease first due to dilution, and then increase because the mobilized contaminants migrate from the source zones to the wells. The highest recharge scenario has the earliest and highest peak in contaminant concentrations due to a stronger remobilization effect, but it has the lowest concentrations and highest pH later due to dilution. In the later period, the increasing recharge again causes dilution due to flushing, resulting in a concentration level below the base case. Because of long-term dilution, the aqueous uranium concentration in greater increasing recharge scenarios is even lower than the base case at the source-zone well after 2035. The relationships between concentrations and recharge are nonlinear and nonmonotonic, depending on different times and locations. Changing recharge rate has less impact at the downgradient well where the spikes are delayed for approximately 10 years since its location is further from the seepage basin, and it takes time for the remobilized plume to reach it. The breakthrough curves of smaller (+10% ~ +30%) increasing recharge scenarios are similar to those of the base case, with slight dilution effects

throughout 2100, while concentration spikes due to remobilization in 2040 are observed with the larger (+40 ~ +50%) increasing recharge scenarios (Figure 6eh).

In the early stage of decreasing recharge scenarios, contaminant concentrations increase because of diminished flushing and a low flow rate of clean groundwater. Later in the simulation period, contaminant concentrations decrease significantly in the greater decreasing recharge scenarios, when the groundwater table declines and isolates the residual contaminants in the vadose zone. This was not observed in the previous tritium simulation (Libera et al., 2019)

because of tritium's radioactive decay. In general, this means that decreasing precipitation and droughts are effective in sequestering contaminants in the vadose zone. At the same time, it implies less flushing and an increase in residence time of the contaminants at the site. The uncontaminated groundwater from the upgradient also migrates more slowly in the aquifer. The larger volume of residual contaminants could potentially increase risk, particularly considering

extreme precipitation events, which are projected to happen more frequently in many climate models (USGCRP, 2017). Also, there is more interest in groundwater resources during a drought, which leads to increased pumping in the contaminated aquifer. Although such pumping activities are strictly regulated at our study site, such trade-offs require attention at other sites.

To investigate the impact on reactive species such as uranium, we compared reactive (uranium) and nonreactive (nitrate) concentration ratios to assess the impacts of reaction and sorption. We originally hypothesized that increasing recharge would decrease reactive species concentration further, since increasing the volume of water in the domain would increase pH, which limits the mobility of uranium. However, Figure 7 shows that the uranium-concentration

ratios compared to the base case increase more significantly than the nitrate concentrations. This is because the remobilization occurs when the pH is still low, and also because remobilization happens to both uranium and protons (Figure 6). In addition, the amount of the residual contaminants is larger for uranium than nitrate due to sorption. Later in the simulation period, the uranium average concentrations are lower than for nitrate and decrease with greater

recharge scenarios, because increasing pH, due to long-term dilution by additional recharge, immobilizes uranium.

        In cap-failure scenarios, sorption of uranium is reduced with increasing infiltration, because $K_d$ is sensitive to lower pH due to remobilization through the basin. At the downgradient well, the

greater recharge cases (+30% ~ +50%) have a more closely correlated $K_d$ and pH, and have a higher aqueous uranium concentration, than the lower recharge scenarios. In our scenarios, there is a clear change in the balance of aqueous and sorbed uranium concentration in the transition from +20% to +30% recharge, where the system's sorption in the downgradient fundamentally changes. The cap-failure cases indicate that changing recharge and cap-failure

levels can trigger dramatic changes in pH and sorption. Similar to Libera et al. (2019), this study confirms the importance of cap or surface barriers to limit the impacts of cap failure under extreme climate regimes. Uniquely, the uncertainty of $K_d$ value was constrained to $10^2$ to $10^3$ in our study, compared to the greater range ($10^2$ - $10^6$) in the previous studies (Bea et al., 2013)

At this site, pH and uranium concentration fronts are retarded, because they are affected by the adsorption and ion exchange processes onto kaolinite and goethite (Bea et al., 2013). Limited sites for adsorption/exchange are saturated by the elevated $H^+$ and uranium loading, and their concentrations eventually reach steady state at the end of the projection period. Overall in our scenarios, the change in recharge has a similar impact on uranium and nonreactive species,

which is largely attributed to pH buffering due to mineral precipitation. The increase in pH due to dilution encourages the precipitation of kaolinite, but the precipitation reaction of kaolinite produces $H^+$ ions, which then decreases pH. At low pH, the hydroxyl groups on the octahedral structures of aluminosilicates like kaolinite become protonated, effectively creating a net positive charge on the mineral. This means that uranium cannot sorb to the clay and is therefore mobile

in the system. Previous experimental (Dong et al. 2012) and modeling (Arora et al., 2018) studies also reaffirmed that percent U(VI) sorption is greater with a higher, neutralized pH, because U(VI) and $H^+$ are competing in sorption. This is the process of dissolution and precipitation of kaolinite:

$$Al_2Si_2O_5(OH)_4 + 6H^+ \leftrightarrow 2Al^{3+} + 2SiO_2 + 5H_2O$$

A similar reaction occurs with gibbsite. Dong et al. (2012) showed that there was an insignificant weight percent or volume fraction of gibbsite at F-Area, since it only forms at pH>5.4. However, in the decreasing recharge scenarios, all the recharge cases at the downgradient well have a pH between 5.4 and 7 after 2070, and pH at the two greater recharge cases at the source-zone well also surpass 5.4. Decreasing recharge would likely trigger the formation of gibbsite, which

could increase pH buffering. Additionally, according to Bea et al. (2013), this mechanism, as well as cation exchange and adsorption processes on kaolinite and goethite, explain some buffering of pH. The pH buffering effect is the major mechanism for pH remaining low for an extended period of time in climate resilience studies with reactive transport modeling. Nevertheless, our model is built upon more than 10 years of site characterization, sorption

experiments and reactive transport models (Dong et al., 2012; Bea et al., 2013; Sassen et al., 2013; Wainwright et al., 2019). The non-electrostatic sorption model (NEM) sorption model used in this study is based on Arora et al. (2018), which is calibrated with long-term monitoring datasets and considered competitive $H^+$ and uranium sorption.

In addition to understanding the impact of a range of recharge scenarios, this study has established a pipeline to use the CMIP5 climate model projections as input to the hydrology and reactive transport modeling simulations. Although increasing precipitation is projected over time, we found that the increasing ET associated with temperature can reduce the recharge rates. We found that, compared to the base case and hypothetical scenarios, the CMIP5 climate data

projects a small increase or no change of recharge rate over time, indicating that the changing climate has minor effects on the contamination plume and breakthrough curves in our study site. This is similar to the behaviors observed in the increasing precipitation scenarios in Figure 6: that smaller recharge increases have little impact on the concentration breakthrough curves, because the increasing recharge is below the threshold that may cause significant

remobilization. Contaminant migration is more controlled by the transport process. Our reactive transport modeling with CMIP5 projection recharge shows that contaminant migration is sensitive to recharge rate. In our study, ET is prescribed from the ensemble average of CMIP5 datasets and is not computed in our simulations. The annual variability of precipitation and ET,

in other words net infiltration, is more significant in RCP8.5 than other scenarios. The
uncertainty in the estimation of ET as well as the annual variability in CMIP5 scenarios could
significantly affect the assessment of waste disposal and contaminant transport.

# 4. CONCLUSION

The climate resilience of residual contamination at the SRS F-Area waste disposal site
throughout the projection period from 2020 to 2100 is investigated in this study. Groundwater
flow, mineral reactions, surface complexation sorption, and ion-exchange processes are
simulated by the Amanzi and PFLOTRAN flow and reactive transport model. We illustrate four
scenarios characterized by a range of variable recharge values: (1) increasing recharge after
2020, (2) decreasing recharge after 2020, (3) cap failure and constant positive recharge shift,
and (4) recharge rate under different RCP scenarios from the CMIP5 climate model projection.
Although exaggerated in the first three cases, this systematic study using changing recharge
rates was useful in identifying the phased impacts of increasing or decreasing recharge rates,
as well as the difference between the reactive and nonreactive species. Plume distribution and
breakthrough curves of chemical species are evaluated to assess the impacts of changing
recharge rate and flow conditions. The ratios of maximum and average reactive and nonreactive
species concentrations between scenarios and base case are used to understand how climate
change affects the adsorption and ion exchange of residual contaminants in the subsurface
domain. Furthermore, $K_d$ breakthrough curves are evaluated to understand the pH effects on
sorption with different recharge rates in those scenarios.

With increasing recharge rates, pH decreases and residual contaminant concentrations
increase, because of the remobilization of protons and reactive species. The impact on uranium
or pH-dependent species is the same as nonreactive contaminants. $K_d$ values are correlated
with pH and behave differently when changing recharge rates beyond certain thresholds. In
most cases, uranium-maximum concentration ratios against the base case are higher than the
nitrate concentration ratios, owing to remobilization, while the uranium concentration
breakthrough curves in the later period depend on long-term flow conditions. The results of cap-
failure scenarios suggest that reactive transport modeling and analysis of pH effects on reactive
species are important for the risk assessment of such engineering failures.

Our results highlight that climate change impacts may not be intuitive, and must be analyzed
quantitatively by models. ET projection has great uncertainty, but is particularly important in
determining the recharge rates in reactive transport modeling for climate resilience studies.
Reactive transport models which consider pH dependency for reactive species are essential for
analyzing the impacts of pH with changing recharge rates. Although this study is focused on one
site, we developed the pipeline to use climate projection datasets in reactive transport modeling
and thereby demonstrated the capability for assessing climate change impacts on waste
disposal sites. We expect that our approach and insights are transferable to other sites that
have large amounts of residual contaminants in the vadose zones or in the groundwater.

**ACKNOWLEDGMENTS**

This study is supported by the Department of Energy, Office of Environmental Management
Technology Development Program under ALTEMIS - Advanced Long-Term Environmental
Monitoring Systems project, the Department of Energy's Savannah River Area Completion
project, as well as by the Office of Science, Biological and Environmental Sciences under
Scientific under the Scientific Focus Area. This research used resources of the National Energy
Research Scientific Computing Center (NERSC), a U.S. Department of Energy Office of
Science User Facility located at Lawrence Berkeley National Laboratory, operated under
Contract No. DE-AC02-05CH11231. This research also used the Lawrencium computational
cluster resource provided by the IT Division at the Lawrence Berkeley National Laboratory
(Supported by the Director, Office of Science, Office of Basic Energy Sciences, of the U.S.
Department of Energy under Contract No. DE-AC02-05CH11231). We appreciate the
comments and suggestions from the handling editor Dr. Brian Berkowitz, reviewer Dr. Jinwoo Im
and an anonymous referee.

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
