# Peer review of "Reactive Transport Modeling for Supporting Climate Resilience at Groundwater Contamination Sites"

_Hydrology and Earth System Sciences, 2021_

## Author Comment (AC1)

Referee #1
**General comments**
This is a very interesting and resourceful paper to read. Authors integrated climate data in hydro-geochemical modeling to investigate climate resilience at groundwater contamination sites under different scenarios. By simulating reactive transport in groundwater, they found what geochemical mechanism plays a major role in uranium transport. The results would help decision-makers to manage the site and prepare for potential risks from climate change. Since this paper overarches from the general concepts (e.g., resilience) to specific mechanisms in modeling (e.g., dilution and remobilization), I would like to ask some questions that help people to have a better understanding of this paper:

Response: Thanks for your feedback and comments. Please find the point-to-point answers below.

**Specific comments**
1) The definition of climate resilience authors made is clear. How would you connect the results to the climate resilience? Are you able to quantify the climate resilience as an environmental metric?  e.g., contaminants' concentrations or pH at an environmentally sensitive location.

Response: The averaged concentrations or pH over time in the climate scenarios at environmentally sensitive locations (in this case, the two monitoring wells) are the environmental metrics for supporting quantifying climate resilience. Resilience is usually quantified as the capability to return back to the system's original condition, in this case the concentrations at monitoring wells or other sensitive locations at the end of the simulation in comparison to the background baseline concentrations.

2) What is the difference between enhanced and monitored natural attenuation for the target contamination site? Do you mean the construction and destruction of the funnel-and-gate system?

Response:  Enhanced natural attenuation is more invasive and tends to be more energy intensive i.e. pump and treat. MNA is more passive, encourages attenuation and relies more on continuous monitoring. They completely stopped with the pump and treat method. This was mostly because it was very energetically demanding and costly/unsustainable.

We are mostly discussing the funnel-and-gate system and barriers, which were constructed and in operation since 2004. We revised this sentence to "The funnel-and-gate system is operating and requires the injection of base solution to increase pH and immobilize contaminants. Quantitative estimation from the modeling study will provide insights for site management when the site transitions to natural attenuation status without any treatments. "

3) I understand that the flow and transport model is well established to describe spatio-temporal evolution of the contaminants of concern. Nevertheless, I am wondering about the limitation of the model as well, e.g., is the sorption model able to capture all sorption mechanisms?

Response: We agree that all the models have some limitations. We would note that our model is built upon more than 10 years of site characterization, sorption experiments and reactive transport models (Dong et al., 2012; Bea et al., 2013; Sassen et al., 2013; Wainwright et al., 2019). Extensive studies have been done focused on sorption experiments and model development (Dong et al., 2012). Our sorption is based on Arora et al. (2018), which developed a non-electrostatic sorption model (NEM) and calibrated geochemical parameters at the SRS site. In the revised manuscript, we will emphasize the past developments contributing to our model, by the following text in Line 639: "The non-electrostatic sorption model (NEM) sorption model used in this study is based on Arora et al. (2018), which is calibrated with long-term monitoring datasets and considered competitive H+ and uranium sorption".

4) The flow and transport model assumed that hydrogeological properties are homogeneous within each unit, and there is no dispersion. However, dispersion could have some impacts when the flow rate is slow, e.g., decreasing recharge scenarios. What impact would you expect on the results if the model considers dispersion due to natural heterogeneity of subsurface (e.g., permeability)?

Response: This issue has been addressed by Bea et al (2013). To clarify, we will include the citation and the following texts: "The system was considered to be advection dominated, so that hydro-mechanical and diffusion transport processes were neglected".

5) Could you explain why there is the nonlinear relationship between recharge and uranium concentrations (around +20~30%?) with specific pH values? You already explained it clearly with specific mechanisms (pH buffering from kaolinite and goethite). However, in decreasing recharge cases, it is much easier to understand because I could compare the pH range of gibbsite formation (>5.4) with simulation results.

Response: We assume the referee is referring to the uranium breakthrough curves in Figure 6d. Due to the different scales of pH and uranium, the non-linear relationship between recharge and uranium is more evident, but pH oscillates as well. This was mentioned in Line 343 and revised to highlight in line 350: "The uranium concentrations (Figure 6d) are also similar to the breakthrough curves for the nitrate concentrations and show negative correlation with pH oscillation."

6) What do you mean by the uncertainty (in line 652)? Do you mean annual variability mentioned in line 543? Is the variability of net infiltration also the same across all climate scenarios?

Response: We mean the uncertainty of both precipitation and ET in CMIP5 projection, which is an ensemble average of multiple climate model outputs. The variabilities of those climate model outputs were not considered. The annual variability of precipitation and ET hence net infiltration are more significant in RCP8.5 than other scenarios.

We revised this sentence as "Our reactive transport modeling with CMIP5 projection recharge shows that the contaminant migration is sensitive to recharge rate. In our study, ET is prescribed from the ensemble average of CMIP5 datasets and is not computed by our Amanzi simulations. The annual variability of precipitation and ET hence net infiltration are more significant in RCP8.5 than other scenarios. The uncertainty of ET estimation as well as the annual variability in CMIP5 scenarios could significantly affect the assessment of waste disposal and contaminant transport."

**Miscellaneous comments**
1) You might want to mention "total recharge" instead of "total runoff" in line 267

Response: Yes, we agree that "total runoff" here can be misleading, and will correct it as "total recharge" in the revision.

2) You can specify ET is evapotranspiration before first mentioning it in line 643

Response: Agree. We change it to "Evapotranspiration (ET)" in Line 643

---

## Author Comment (AC2)

Referee #2

Review of HESS-2021-338-2: Reactive transport modeling for supporting climate resilience at groundwater contamination sites

This paper reports an interesting coupling of climate projection data with subsurface reactive transport simulations through the Amanzi platform. The study is developed in application to the Savannah River site F area and focuses on the effects of perturbation scenarios on recharge and the distribution of uranium and nitrate concentrations. The key result is that pH is impacted by dilution and remobilization, which influence the sorption of uranium onto sediments. Simply for the novelty of this method and the effort to offer projections into future climate scenarios, this study could represent an advancement in the field. However, I do think there are several areas that require strengthening to lend fidelity to these model results.

Response: Thanks for your feedback and comments. Please find the point-to-point answers below.

Major edits:

1. Testable hypotheses: Line 73 refers to testable hypotheses, but I see no such statement offered in the text to this point. It reads like there is a lot of competing information and many factors in play. This is not a testable hypothesis. This needs to be carefully revised – offering a clear and testable statement would greatly strengthen the purpose and scope of the study

Response: We revised the hypothesis explicitly to "We hypothesized that increasing recharge would decrease reactive species concentration further, since increasing the volume of water in the domain would increase pH, which limits the mobility of uranium."

2. Vegetation: there seems to be no treatment of the role vegetation plays in recharge and near-surface water storage. Everything is limited to assuming that the effects of changing precipitation can be emulated by changes in recharge rate. This would seem to undermine the coupling of these climate models and thus the overall impact of the study

Response: We agree that vegetation plays a key role controlling ET and water budget. Although the land models are available for plant processes, these models are typically computationally intensive. To accommodate the variable impacts of vegetation, ET as well as precipitation, we have varied the recharge values which are the combined effects of the precipitation and ET. In particular, our groundwater is mostly deeper than the root zone except for the region close to the seepage so that we can consider the recharge or net infiltration below the root zone as near surface processes (a thin layer compared to the overall domain depth of 50 meters). By changing the recharge values, we can explore how this combined impact affects the residual contaminants. In addition, most climate projections provide precipitation and ET at a given location. Considering a recharge or net infiltration will be appropriate for a groundwater model; without including computationally expensive land process models. We will change these texts in the revised manuscript: "We assume that the recharge represents the combined impacts of precipitation and ET. This treatment would be appropriate for this domain and most groundwater models in which the groundwater domain is deep compared to the root zone depths"

3. Prior work: there seems to be a lot of overlap with Libera et al (2019) and Bea et al. (2013) with regard to the reactive transport simulations. Figure 2 seems to be largely reproduced from Libera. The distinction between these models and prior work should be clearly explained. Presently I am left with the sense that this paper is a melding of Bea et al. (2013) + Libera et al. (2019) + the climate scenarios. Perhaps this is enough to argue that the study is novel, if so, this should be explicitly detailed. Further, it is unclear why this particular location was chosen for the purposes of such a model – is it because the Libera et al. paper already existed or is there some stronger reason why this is the appropriate location to work on the Amanzi development?

Response: In our previous studies, Bea et al. (2013) used another reactive transport model (TOUGHREACT) with non-electrostatic model. Libera et al. (2019) used the Amanzi model to simulate flow and tritium transport under various climate scenarios. In our study, we used the Amanzi model to simulate the full chemistry with mineral reactions and sorption processes, and explicitly highlighted the effects on pH and uranium. The flow part is the same as Libera et al. (2019). We agree that the novelty of reactive transport modeling should be highlighted in the revision. We explicitly add the demonstrations and revise the text in Line 72: "However, Libera et al. (2019) only simulated tritium, and does not couple with a reactive transport model for simulating other chemical species, sorption, and mineral reactions."

This particular location was chosen because of the historic contamination monitoring activities in the SRS F-Area seepage basin, and the series of previous modeling studies (Bea et al., 2013, Libera et al. 2019, etc) that existed. It has extensive characterization and model development. The vadose zone residual contaminants are quite common (e.g., Stubbs et al., 2009, Zachara et al., 2005) and also the contaminant export through the wetland region are fairly common features across many contaminated sites as well (e.g., Mansoor et al., 2006; Li et al., 2014, Change et al., 2014). We revised the text in Line 88: "The SRS F-Area seepage basin was chosen because of the historic contamination monitoring activities, and extensive characterization and model development (e.g., Flach, 2004; Bea et al., 2013; Sassen et al., 2012; Wainwright et al., 2014, 2015, 2016; Denham and Eddy-Dilek, 2017; Libera et al., 2019). More importantly, the contamination in F-Area is representative that our study may provide broader insights to other contamination sites. The vadose zone residual contaminants are quite common (e.g., Stubbs et al., 2009, Zachara et al., 2005) and also the contaminant export through the wetland region are fairly common features across many contaminated sites as well (e.g., Mansoor et al., 2006; Li et al., 2014, Change et al., 2014)."

4. Advection dominated: the authors are well aware of how profoundly important the effects of diffusion and dispersion are when dealing with sorption and solute exchange between high and low flow zones. The assumption that this system is advection dominated, along with the large uncertainty in Kd values, would seem to place significant uncertainty on the present results. This assumption must be clearly explained and justified. At present it seems to be simply stated in section 3.2 without further consideration

Response: At the scale of our F-Area region (< 1000 m domain), the contaminant spreading will be typically dominated by advection. We will highlight that the system is advection dominated in the model description by revised the text in Line 205: "Based on the study of scale-dependent advection and dispersion processes in Molz (2015) that contaminant transport will be typically dominated by advection at scale of 1000 meter, the system considered to be advection dominated, and mechanical dispersion and molecular diffusion transport processes are neglected"

The Kd uncertainty ranged 10e2 to 10e6 in the past studies, but has been narrowed down to 10e2 to 10e3 in our study. We revised the text in Line 672 as "The uncertainty of Kd value was constrained to 10e2 to 10e3 in our study, compared to the greater range (10e2 - 10e6) in the previous studies (Bea et al., 2013)".

5. pCO2: Table 2 reports CO2(g) concentrations that appear to be lower than even present atmospheric values, and certainly do not appear to consider changes in pCO2 associated with a changing climate. Why isn't this considered in the model along with recharge variability?

Response: We used the same pCO2 concentration as previous publications (Bea et al., 2013). We do not consider change in pCO2 associated with a changing climate and agree that this is an uncertainty to discuss. First, increasing pCO2 could decrease recharge pH but the effects are relatively small (the acid rain with pH < 7 is not because of pCO2, but mostly sulfuric or nitric acid). Also, increasing temperature may decrease CO2 solubilities in the water. We will add the text in Line 226: "The pCO2 concentrations are based on previous publications (Bea et al., 2013) and assumed constant over the simulation, as pCO2 concentration does not significantly affect pH".

Specific edits:
L40: "significant amounts" clarify what this means
Response: Rephrase as "expanded contamination plume with high concentration of tritium, uranium and other chemical species"

L51: what is meant by "absorb the projected stresses"
Response: Rephrase as "to be affected by"

L54: this is circular. First the authors argue that we don't know how climate changes and associated stresses may impact contaminated sites, then it states that this information is a critical need. How can it be both unknown and critical?
Response: We don't have the quantitative estimation of climate change impacts, and don't fully understand the process of how climate change could impact contaminated sites. However, we know that understanding the changing climate impacts could be critical, from various previous studies and raising the concerns from the important stakeholders. We revised this sentence as "A critical need exists for understanding climate change impacts on contaminated sites (e.g.,

U.S. EPA, 2014 and DOE, 2017), however, a quantitative estimation with climate change projection is still missing."

L55-65: please revise this text. It's quite confused and hard to follow
Response: We will revise as "Evaluating the effect of climate change on the abundance of water resources has been widely studied (e.g., Gellens and Roulin, 1998; Green et al., 2011; Middelkoop et al., 2001; Pfister et al., 2004), however, water quality and contamination issues were less investigated (Visser et al., 2012). Most previous researches study surface water (Wilby et al., 2006; Van Vliet and Zwolsman, 2008; Van Bokhoven, 2006; Futter et al., 2009; Schiedek et al., 2007), because of the accessibility and data availability (Green et al., 2011). In the limited studies for climate change impacts on groundwater in the subsurface domain, agricultural effluents at the regional scale are the research focus (Bloomfield et al., 2006; Futter et al., 2009; Li and Merchant, 2013; Olesen et al., 2007; Sjoeng et al., 2009; Whitehead et al., 2009; Wilby et al., 2006; Darracq et al., 2005; Destouni and Darracq, 2009; Park et al., 2010). "

L70-71: "appear in different phases" what does this mean?
Response: The "phases" was a term in Libera et al (2019) for different periods of waste discharge or residual contamination. Rephase as "before and after the changing precipitation"

L73: tritium is non reactive?
Response: Tritium decays but does not react with minerals or participates in sorption. Therefore, simulating tritium does not require the coupling with a reactive transport model. We will revise this sentence as "Libera et al. (2019) only simulated tritium, which decays but does not couples with a reactive transport model and do simulate the mineral reaction and sorption"

L133: that's a huge range of variability in a key parameter (Kd). How is this much uncertainty accommodated in the model?
Response: We didn't use Kd as an input parameter in our reactive transport model. Instead, we compute Kd in the results section and show that our Kd output ranges between 10e2 to 10e3. As mentioned in response to referee #1, we revised the text in Line 672 as "The uncertainty of Kd value was constrained to 10e2 to 10e3 in our study, compared to the greater range (10e2 - 10e6) in the previous studies (Bea et al., 2013)".

L143: state the criteria for transition from enhanced to monitored natural attenuation
Response:  Enhanced natural attenuation is more invasive and tends to be more energy intensive i.e. pump and treat. MNA is more passive, encourages attenuation and relies more on continuous monitoring. They completely stopped with the pump and treat method. This was mostly because it was very energetically demanding and costly/unsustainable.  We agree this sentence can be misleading and revised this sentence as "The funnel-and-gate system is operating and requires the injection of base solution to increase pH and immobilize contaminants. The quantitative estimation from the modeling study will provide insights for site management when the site transitions to natural attenuation status without any treatments. "

L179: This seems to strongly overlap with the Libera et al. study

Response: It is true that the flow simulations are the same as Libera et al. (2019). However, this study focuses on the assessment of chemical species, while Libera et al. (2019) only simulated flow and tritium transport. We explicitly add the demonstrations and revise the text in Line 72: "However, Libera et al. (2019) only simulated tritium, and does not couple with a reactive transport model for simulating other chemical species, sorption and mineral reactions."

L197: state the model used by Bea et al. (2013)
Response: We add the text as "Our Amanzi simulation used the same conditions of mineral composition and kinetic reactions as the TOUGHREACT model in Bea et al. (2013)."

L221: kinetic rate constant. Kaolinite is not a primary mineral
Response: Agree and change kaolinite is not a primary mineral.

L266-269: does this undermine the study?
Response: The goal of this paper is not just prediction itself but also understanding the impact of increasing and decreasing recharge on non-reactive vs reactive contaminants. Scenario-based studies are useful for such understanding. In parallel, having this more realistic scenario is to demonstrate the actual implementation of models at the particular site. The fact that there is no significant impact observed is a valid result for this site, but the understanding would be more transferable to other sites as well.

Although changing recharge rates are not significant in the projection, simulation with climate scenarios are important in long-term robustness of contamination remediation activities in the contaminated sites. We revise in L266 and add the explanation: "Although total runoff only slightly increases as both precipitation and evapotranspiration are increasing (hence the difference is offset), our simulations focus on gaining the understanding and quantitative estimation of changing climate impacts on the long-term robustness of contamination remediation in the F-Area."

L288: this cap failure scenario seems very fictitious and may only be included to essentially get the model to do something noticeable. I'm not convinced this is a strong addition to the study
Response: Yes it is fictitious, but not impossible. Multiple previous studies have either investigated the possibility or documented the impacts of such damage events in contamination sites. We add the text in Line 288: "Multiple studies have demonstrated that increased vegetation or other mechanisms could threaten or completely damage the integrity of the source-zone capping structure (Worthy et al., 2013a, 2013b, 2015)."

Figure 5: are these data supposed to suggest model fidelity? The fits appear quite poor.
Response: We have used the same parameters as Bea et al (2013), and had a similar fit to the observations as Bea et al (2013). Also, the objective of this study focuses on evaluating the responses to changing precipitation and other conditions in the projection, and does not focus on the assessment in the historical period.

References:

Molz, Fred. "Advection, dispersion, and confusion." Groundwater 53.3 (2015): 348-353.

U.S. DOE, "Climate change vulnerability screenings", (2017)

U.S. EPA, "U.S. Environmental Protection Agency Climate Change Adaptation Plain", (2014), EPA 100-K-14-001

Stubbs, J. E., Veblen, L. A., Elbert, D. C., Zachara, J. M., Davis, J. A., & Veblen, D. R. (2009). Newly recognized hosts for uranium in the Hanford Site vadose zone. Geochimica et Cosmochimica Acta, 73(6), 1563-1576.

Zachara, J. M., Davis, J. A., Mckinley, J., Wellman, D. M., Liu, C., Qafoku, N., & Yabusaki, S. B. (2005). Uranium geochemistry in vadose zone and aquifer sediments from the 300 Area uranium plume.

Chang, H. S., Buettner, S. W., Seaman, J. C., Jaffé, P. R., Koster van Groos, P. G., Li, D., ... & Kaplan, D. I. (2014). Uranium immobilization in an iron-rich rhizosphere of a native wetland plant from the Savannah River Site under reducing conditions. Environmental science & technology, 48(16), 9270-9278.

Li, D., Seaman, J. C., Chang, H. S., Jaffe, P. R., van Groos, P. K., Jiang, D. T., ... & Kaplan, D. I. (2014). Retention and chemical speciation of uranium in an oxidized wetland sediment from the Savannah River Site. Journal of environmental radioactivity, 131, 40-46.

Mansoor, N., Slater, L., Artigas, F., & Auken, E. (2006). High-resolution geophysical characterization of shallow-water wetlands. Geophysics, 71(4), B101-B109.